# Overweight and obesity association with mortality in patients with heart failure and reduced or preserved ejection fraction-a cohort study

Nicolas Garin[1,2,3‡]*, David Carballo[3,4‡], Jonathan Dash[2‡], Jérôme Stirnemann[2,3‡], Jean-Luc Reny[2,3‡], Nicolas Vuilleumier[3,5‡], Sebastian Carballo[2,3‡]

1 Division of Internal Medicine, Riviera Chablais Hospital, Rennaz, Switzerland, 2 Division of General Internal Medicine, Geneva University Hospitals, Geneva, Switzerland, 3 Faculty of Medicine, Geneva University, Geneva, Switzerland, 4 Division of Cardiology, Geneva University Hospitals, Geneva, Switzerland, 5 Division of Laboratory Medicine, Department of Diagnostics, Geneva University Hospitals, Geneva, Switzerland

‡ This author takes responsibility for all aspects of the reliability and freedom from bias of the data presented and their discussed interpretation.
* Nicolas.garin@hug.ch

## Abstract

### Background

Obesity is a risk factor for incident heart failure, but patients with excess weight and heart failure have lower mortality. This "obesity paradox" may be explained either by a favourable effect of the adipose tissue or by confounding.

We aimed to assess if body mass index (BMI) is associated with lower mortality after extensive adjustment for prognostic factors in patients with reduced (HFrEF) or preserved (HFpEF) ejection fraction.

### Methods

Prospective, observational study including consecutive patients hospitalized for acute heart failure. Two years hazard of mortality was assessed in a multivariable Cox model, in the whole population and separately for HFrEF and HFpEF.

### Results

Among 957 included patients (mean age 76 years, 41% women), 500 (47%) had HFrEF and 443 (53%) HFpEF. Four hundred (39%) were in the normoweight, 301(30%) in the overweight, and 256(25%) in the obese category (Class I obesity:144 patients; class II or III: 112). Corresponding mortality was 37%, 26% and 22%. Unadjusted hazard ratio (HR) for mortality was 0.96 (95% CI 0.94–0.98) for each BMI point in the whole population, 0.97 (95% CI 0.94–1.02) in patients with HFrEF, and 0.94 (95% CI 0.92–0.97) in HFpEF. After adjustment for age, sex, atrial

**Data availability statement:** All relevant data are within the paper and its Supporting Information files.

**Funding:** The author(s) received no specific funding for this work.

**Competing interests:** The authors have declared that no competing interests exist.

fibrillation, diabetes, chronic obstructive pulmonary disease, chronic anaemia, hypertension, glomerular filtration rate, and NT-proBNP, HR was 1.00 (95% CI 0.96–1.02) in the whole population, 1.02 (0.96–1.07) in HFrEF, and 0.98 (95% CI 0.94–1.01) in HFpEF.

## Conclusions

Excess weight was associated with an apparent survival benefit in patients with acute heart failure, particularly in patients with HFpEF. This advantage disappeared completely after adjustment for confounding factors including NT-proBNP. The obesity paradox can be completely explained by differences in demographics, co-morbidities, and severity of heart failure.

## Introduction

Obesity and overweight are increasingly present in countries both with higher and lower socio-demographic index and are associated with chronic health conditions (diabetes, hypertension) and adverse outcomes [1]. Obese patients are at increased risk of developing heart failure [2,3]. However, once heart failure is diagnosed, obese patients have a lower risk of mortality, the so-called "obesity paradox", already described 20 years ago [4,5]. This unexpected association has been consistently described across the world, in chronic and acute heart failure, in heart failure with preserved (HFpEF) or reduced (HFrEF) ejection fraction, and is already apparent in the short term [4–7]. Causal hypotheses to explain the obesity paradox include a protective effect of excess weight against cachexia induced by heart failure, or attenuation of pathogenic pathways by the adipose tissue. Non-causal hypotheses include confounding by multiple factors (as obese patients are generally younger, have a higher left ventricular ejection fraction, a higher blood pressure allowing for up titration of disease-modifying drugs) [8]; or lead-time bias, i.e., obese patients with heart failure might be diagnosed at an earlier stage in the evolution of the disease because they present earlier with symptomatic impairment [9,10].

Because of this uncertainty, international guidelines for heart failure treatment lack clear recommendations about weight management in overweight and obese patients [11,12]. As recently proposed treatments for symptomatic heart failure in obese patients lead to substantial weight loss [13–15], understanding the intertwined relations between obesity and prognosis in heart failure becomes increasingly pressing. We aimed to test if body mass index (BMI) remained an independent predictor of the risk of death in patients hospitalized for acute heart failure after extensive adjustment for confounders, and if the relation differed for patients with HFpEF or HFrEF.

## Methods

### Population

Consecutive adult patients admitted for acute heart failure between 1st of November 2014 and 30th of October 2021 either acutely decompensated or de novo, at Geneva

University Hospitals were included in a registry (ClinicalTrials.gov: NCT02444416). All patients gave written informed consent. Inclusion required the presence of symptoms and signs suggestive of heart failure according to the European Society of Cardiology definition, along with elevated natriuretic peptides (B-type natriuretic peptide (BNP)>100 pg/ml or N-terminal pro-B-type natriuretic peptide (NT-proBNP) >300 pg/ml) [11]. All inclusions were reviewed by a senior investigator, expert in the management of heart failure. Data collection included demographic characteristics, co-morbidities, symptoms and signs at admission, vital parameters, and an extensive panel of blood tests. Echocardiography was mandatory for all patients. Data concerning the hospitalization (length of stay, medications use, complications) were also recorded. Outcomes were prospectively collected at 3, 12, and 24 months and yearly afterwards by tracking readmissions in the hospital electronic medical record, contact with treating physicians, or both, and included mortality and readmission (all-cause, or heart failure-related). The study complied with the Declaration of Helsinki. The protocol for the registry was approved by the institutional ethics committee (CER 14–019).

## Variables and definitions

Body mass index (BMI) was computed as weight (kgs) divided by height $^2$ ($m^2$). Height was self-reported, while weight was measured at admission using a weighting chair as part of routine clinical care. Patients were stratified into four categories, as proposed by the World Health Organization (underweight: BMI<= 18.5kg/m 2; normal:>18.5–25.0kg/m2; overweight>25.0–30kg/m2; obese>30kg/m2). Obese patients were further stratified between class I (BMI 30–35kg/m2) and class II and III (BMI>35kg/m2).

Patients were stratified according to left ventricular ejection fraction (LVEF). Patients with a LVEF >= 50% had heart failure with preserved EF (HFpEF). Patients with a LVEF <40% had heart failure with reduced ejection fraction (HFrEF), in accordance with the definitions of the European Society of Cardiology [11]. Patients with a LVEF between 41% and 49% (mildly reduced ejection fraction) share more characteristics with patients with HFrEF than HFpEF, being younger, more frequently males, with more coronary artery disease and less atrial fibrillation [16]. They were merged with the HFrEF population for the purpose of the present analysis.

## Study outcomes

The primary outcome was all cause mortality at two years. The main secondary outcome was heart failure-related mortality at two years. The primary analysis was a survival analysis, with survival time calculated from the first day of hospitalization.

## Statistical analysis

Descriptive statistics used frequencies with proportions for categorical data and mean with standard deviation (SD) or median with interquartile range (IQR) for continuous data, as appropriate. NT-proBNP was log-transformed, as its distribution was right-skewed. Patients stratified by BMI category were compared with Chi-square or Fisher's exact test (categorical) or Analysis of Variance (continuous). Patients in the underweight category may differ systematically from normal- or overweight patients, as they may be in a chronic catabolic state triggered by active cancer, chronic obstructive pulmonary disease, or frailty. They were excluded from the rest of the analysis.

The association between BMI (as a categorical variable) and the hazard of death was plotted in a Kaplan Meier curve and assessed as a continuous variable in a univariable Cox proportional model. We then adjusted in a multivariable Cox proportional model for age, sex, and for co-morbidities known to be associated with mortality in heart failure. NT-proBNP was finally added to the multivariable model. The analysis was repeated for the secondary outcome. As HFpEF and HFrEF likely have different physiopathology, we repeated the primary analysis separately for patients with HFpEF and HFrEF. Backward conditional selection was used with the same initial variables as a sensitivity analysis. Confidence intervals for the multivariate analyses were estimated with the use of unstratified bootstrapping (1000 samples with replacement).

NT-proBNP is strongly associated both with the risk of death and, inversely, with BMI. We computed the area under the receiver operating characteristic (AUROC) curve for both variables and assessed their correlation with Spearman's rank correlation coefficient.

Age is also strongly associated both with the risk of death and inversely with BMI. As a sensitivity analysis, we tested both variables together in a Cox proportional model, first with BMI as a continuous variable, then with categories of BMI. The proportional hazards assumption was tested by direct examination of the log-minus-log plots.

We did not impute missing data, as they were few, and present complete case analysis. All results are reported with 95% confidence intervals. A p value < 0.05 was deemed significant. No adjustment was done for multiple testing. All analyses were conducted with SPSS version 25 (IBM inc).

## Results

A total of 1020 patients (586 men, 58%) were included in the registry. Mean age was 76 years (SD 14). Four hundred patients (39%) had a normal BMI, 63 (6%) were underweight, 301 (30%) were overweight, and 256 (25%) were obese. Mean BMI was 26.7 kg/ m2 (SD 6.4). Median BMI in the obese category was 34 kg/m2 (IQR 32–37). Class I obesity was present in 144 patients (14%) and class II or III in 112 (11%).

Characteristics of the patients stratified by BMI class are displayed in Table 1. Patients in the underweight category were older, more likely to be women, to have chronic obstructive pulmonary disease (COPD) or chronic anaemia, but less likely to have diabetes, coronary artery disease or chronic renal failure. They also had higher C-reactive protein.

After excluding underweight patients, 957 patients remained in the study. Five hundred (52%) had HFpEF, and 443 (46%) had HFrEF (the information was missing in 14). Obese patients were significantly younger, more likely to have diabetes, hypertension, and COPD, and less likely to have anaemia than normoweight patients. Upon admission, they presented more frequently with lower limb oedema. NT-proBNP was lower, and they were more likely to have HFpEF. They were in similar New York Heart Association (NYHA) class, and the proportion of patients admitted the preceding year was similar than for their normoweight counterpart. Disease-modifying drugs (angiotensin converting enzyme inhibitors, angiotensin receptor blockers or mineralocorticoid receptor antagonists) prescription did not differ according to weight category. SGLT-2 inhibitors prescription was uncommon at the time of this study, and their use was not documented.

In general, overweight patients had characteristics intermediate between normoweight and obese patients.

At the end of the 2 years of follow-up, overall mortality was 29.3%. Mean follow-up was 592 days (95%CI 576−607). All-cause mortality decreased from 37% in patients with a normal weight, to 17% in patients with class II-III obesity. Results were similar for heart failure-related mortality (Table 2). The hazard of death according to BMI category differed significantly on the Kaplan Meier curves (p < 0.001 by logrank) (Fig 1 and S1 Fig). The crude Hazard ratio (HR) for mortality was 0.96 (95% CI 0.94–0.98) with each additional BMI point, confirming the presence of an obesity paradox in the cohort.

In a first multivariate model adjusting for age, sex, atrial fibrillation, diabetes, COPD, chronic anaemia, hypertension, and glomerular filtration rate (GFR), the adjusted HR (aHR) was 0.97 (Table 3). However, after adding NT-proBNP in the model, the aHR increased to 1.00 (95% CI 0.96–1.02) (Table 3).

Heart failure-related mortality at 2 years was significantly lower for obese patients (9%) compared with overweight (17%) or normoweight patients (16%). Crude HR was 0.95 (95% CI 0.92–0.98) and increased to 1.00 (95% CI 0.96–1.04) in the second model including NT-proBNP. Age, chronic obstructive pulmonary disease, and NT-proBNP were independently associated with both all-cause and HF-related death. Chronic anaemia and atrial fibrillation were additional predictors of all-cause mortality.

Heterogeneity in the association between BMI and mortality was present in stratified analysis for HFpEF and HFrEF. The association was not significant for patients with HFrEF, both in unadjusted and adjusted models. Conversely, BMI was strongly associated with the hazard of mortality for patients with HFpEF in the univariate analysis and in the first multivariate model (HR 0.94, 95% CI 0.92–0.97 and aHR 0.95, 95%CI 0.91–0.99). However, the association was no more significant when NT-proBNP was added in the second model (HR 0.98, 95% CI 0.94–1.01) (Fig 2 and 3 and Table 4). Applying

**Table 1. Characteristics of the patients stratified by BMI categories.**

| Number(%) | Underweight <18.5 kg/m² (N = 63) | Normal >=18.5–25 kg/m²(N = 400) | Overweight >=25–30 kg/m²(N = 301) | Obese >=30 kg/m² (N = 256) | p value (overall) | p value (without underweight) |
|---|---|---|---|---|---|---|
| Age, mean (SD), (years) | 79.8(11.3) | 78.8(13.6) | 74.7(14.6) | 71.3(13.3) | <0.001 | <0.001 |
| Sex: …female <br> male | 43(68) <br> 20(32) | 181(45) <br> 219(55) | 98(33) <br> 203(67) | 110(43) <br> 146(57) | <0.001 | 0.002 |
| *Co-morbidities and associated conditions* | | | | | | |
| Diabetes | 3(5) | 94(24) | 107(36) | 134(52) | <0.001 | <0.001 |
| Hypertension | 41(65) | 279(70) | 238(79) | 220(86) | <0.001 | <0.001 |
| Coronary artery disease | 9(14) | 136(34) | 118(39) | 90(35) | 0.008 | 0.24 |
| Chronic renal failure | 16(25) | 131(33) | 104(35) | 90(35) | 0.32 | 0.86 |
| Chronic obstructive pulmonary disease | 15(24) | 47(12) | 34(11) | 46(18) | 0.07 | 0.04 |
| Atrial fibrillation | 27(43) | 191(48) | 130(43) | 117(46) | 0.88 | 0.50 |
| Chronic anaemia | 29(46) | 165(41) | 132(44) | 83(32) | 0.07 | 0.01 |
| Hospitalized last 12 months | 17(27) | 101(26) | 66(22) | 56(22) | 0.57 | 0.45 |
| *Echocardiographic data (missing: 15)* | | | | | | |
| *Ejection fraction* | | | | | <0.001 | <0.001 |
| reduced | 29(47) | 214(54) | 140(47) | 89(35) | | |
| preserved | 33(53) | 183(46) | 155(53) | 162(65) | | |
| *Clinical characteristics at admission (missing: 37)* | | | | | | |
| Heart rate mean (SD) | 87(20) | 93(26) | 91(27) | 94(48) | 0.39 | 0.33 |
| Respiratory rate mean (SD), | 23(6) | 24(7) | 25(8) | 25(8) | 0.49 | 0.59 |
| SBP mean (SD), mmHg | 136(26) | 142(26) | 142(26) | 145(27) | 0.10 | 0.24 |
| DBP mean (SD), mmHg | 76(17) | 84(19) | 83(20) | 83(20) | 0.04 | 0.97 |
| Elevated jugular pressure: ≥ moderate[a] | 4(9) | 32(10) | 23(10) | 12(7) | 0.61 | 0.41 |
| Lower limb oedema: ≥ moderate[a] | 19(31) | 132(33) | 112(38) | 121(48) | 0.002 | 0.001 |
| Rales: ≥ moderate[a] | 12(19) | 94(24) | 84(28) | 70(28) | 0.33 | 0.35 |
| *Dyspnoea according to New York Heart Association class* | | | | | | |
| | | | | | 0.11 | 0.25 |
| …….1-2 | 10(15) | 29(7) | 29(9) | 13(5) | | |
| ……..3 | 18(29) | 158(40) | 111(37) | 93(36) | | |
| ……..4 | 35(56) | 213(53) | 161(54) | 150(59) | | |
| *Laboratory values* | | | | | | |
| NT-proBNP, mean (SD), pg/ml[b] | 11200 (13500) | 10700 (12000) | 7000 (9200) | 4500 (7500) | <0.001 | <0.001 |
| eGFR mean (SD), ml/mn | 56 (26) | 52 (23) | 53 (24) | 56 (26) | 0.13 | 0.09 |
| C-reactive protein, mean (SD), mg/L | 45 (79) | 28 (47) | 31 (53) | 34 (58) | 0.11 | 0.33 |
| Total cholesterol, mean (SD), mmol/L | 4.0 (1.1) | 3.8 (1.1) | 3.8 (1.1) | 3.9 (1.2) | 0.23 | 0.35 |
| *Treatment at discharge* | | | | | | |
| ACE inhibitor or ARB | 37 (62) | 261 (70) | 199 (68) | 178 (74) | 0.27 | 0.39 |
| Betablockers | 40 (66) | 276 (73) | 212 (73) | 172 (72) | 0.64 | 0.93 |
| MRA | 9 (15) | 78 (20) | 67 (23) | 43 (17) | 0.28 | 0.25 |

[a]on a 4 levels ordinal scale (absent; slight; moderate; marked).

[b]includes 123 patients with BNP converted to NT-proBNP using a conversion factor of *6.25(40).

SBP systolic blood pressureDBP diastolic blood pressureeGFR estimated glomerular filtration rate ACE angiotensin converting enzymeARB angiotensin receptor blockersMRA Mineralocorticoid receptor antagonists.

**Table 2. All-cause and heart failure-related risk of death, stratified by BMI categories.**

| Number (%) | < 18.5 (N = 63) | >=18.5–25 (N = 400) | >25–30 (N = 301) | >30-35 (N = 144) | >35 (N=112) | p value |
|---|---|---|---|---|---|---|
| 2-years risk of death (all cause) | 28 (44) | 146(37) | 79(26) | 36(25) | 19(17) | <0.001 |
| 2-years risk of death (heart failure-related) | 16 (25) | 68(17) | 49(16) | 17(12) | 6(5) | 0.003 |
| 2-years risk of death in HFrEF (all cause) | 14 (48) | 62(29) | 29(21) | 14(25) | 5(16) | 0.01 |
| 2-years risk of death in HFrEF (heart failure- related) | 8 (28) | 29(14) | 19(14) | 6(11) | 1(3) | 0.06 |
| 2-years risk of death in HFpEF (all cause) | 13 (39) | 83(45) | 48(31) | 21(25) | 14(18) | <0.001 |
| 2-years risk of death in HFpEF (heart failure- related) | 8 (24) | 38(21) | 28(18) | 10(12) | 5(6) | 0.02 |

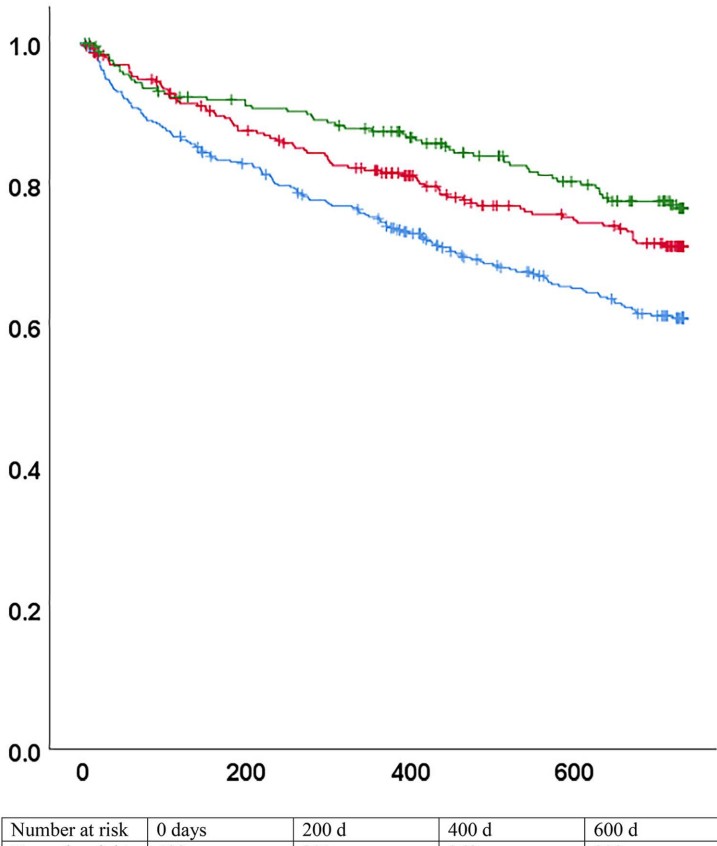

| Number at risk | 0 days | 200 d | 400 d | 600 d |
|---|---|---|---|---|
| Normal weight | 400 | 321 | 268 | 220 |
| Overweight | 301 | 251 | 213 | 184 |
| Obese | 256 | 224 | 203 | 175 |

**Fig 1. Survival by BMI categories, all patients. p < 0.001 (logrank).** Blue: normal weight (BMI 18.5-24.9). Red: overweight (BMI 25-29.9). Green: obese (BMI >= 30). X axis: days since inclusion. Y axis: survival without death.

backward conditional selection in the multivariate models led to the same results. Direct examination of the log-minus-log plots confirmed that the proportional hazards assumption was met.

BMI and NT-proBNP were strongly correlated (Sperman's rho −0.38, p < 0.001). AUROC for 2-years all-cause mortality was 0.58 (95% CI 0.55–0.62) for BMI and 0.64 (95% CI 0.60–0.68) for NT-proBNP, meaning that NT-proBNP could better discriminate mortality than BMI.

**Table 3. Hazard ratio for death.**

| Co-variate | HR or aHR (95%CI) | p value | HR or aHR (95%CI) | p value |
|---|---|---|---|---|
| | 2-years death (all cause) | | 2-years death (heart failure-related) | |
| **Unadjusted** | | | | |
| BMI (kg/m2) | 0.96 (0.94-0.98) | <0.001 | 0.95 (0.92-0.98) | < 0.01 |
| **First model** | | | | |
| BMI (kg/m2) | 0.97 (0.95-1.00) | 0.05 | 0.97 (0.93-1.01) | 0.13 |
| Age (year) | 1.02 (1.01-1.04) | < 0.01 | 1.05 (1.02-1.08) | <0.01 |
| Sex (ref: female) | 0.96 (0.74-1.28) | 0.77 | 0.88 (0.59-1.27) | 0.49 |
| Diabetes | 1.05 (0.79-1.42) | 0.71 | 1.03 (0.66-1.55) | 0.89 |
| Hypertension | 0.82 (0.60-1.19) | 0.23 | 1.24 (0.78-2.18) | 0.41 |
| Chronic obstructive pulmonary disease | 2.17 (1.59-2.99) | <0.01 | 2.74 (1.81-4.14) | <0.01 |
| Atrial fibrillation | 1.28 (0.98-1.67) | 0.06 | 1.39 (1.00-2.06) | 0.07 |
| Chronic anaemia | 1.53 (1.16-1.98) | <0.01 | 1.30 (0.87-1.97) | 0.18 |
| GFR (ml/mn) | 0.99 (0.98-1.00) | 0.01 | 0.99 (0.98-1.00) | 0.01 |
| **Second model** | | | | |
| BMI (kg/m2) | 1.00 (0.96-1.02) | 0.90 | 1.00 (0.96-1.04) | 1.00 |
| Age (year) | 1.03 (1.01-1.04) | <0.01 | 1.05 (1.03-1.08) | <0.01 |
| Sex (ref: female) | 0.98 (0.74-1.27) | 0.86 | 0.89 (0.62-1.26) | 0.52 |
| Diabetes | 1.09 (0.82-1.47) | 0.60 | 1.06 (0.68-1.64) | 0.79 |
| Hypertension | 0.77 (0.57-1.05) | 0.09 | 1.13 (0.69-2.01) | 0.63 |
| Chronic obstructive pulmonary disease | 2.19 (1.60-3.03) | <0.01 | 2.76 (1.74-4.23) | <0.01 |
| Atrial fibrillation | 1.30 (1.02-1.70) | 0.04 | 1.42 (0.97-2.08) | 0.07 |
| Chronic anaemia | 1.49 (1.11-1.98) | 0.01 | 1.26 (0.83-1.82) | 0.25 |
| eGFR (ml/mn) | 1.00 (0.99-1.00) | 0.44 | 1.00 (0.98-1.00) | 0.32 |
| NT-proBNP (log-transformed) | 2.16(1.50-3.06) | 0.01 | 2.45 (1.59-4.05) | <0.01 |

BMI was an independent predictor of 2-years mortality when analysed together with age, both as a continuous variable (aHR 0.97, 95% CI 0.95–0.99) and in category (aHR 0.75 [95% CI 0.57–0.99] for overweight vs. normoweight; aHR 0.64 [95% CI 0.46–0.88] for obese vs. overweight).

## Discussion

In this cohort of hospitalized patients with acute heart failure, a higher BMI was strongly associated with a lower risk of mortality, both all-cause and heart-failure related. However, the association was attenuated when adjusting for age and comorbidities and disappeared when NT-proBNP was added to the model. The obesity paradox was absent in HFrEF even in the univariate analysis. It was present in HFpEF and remained associated with lower mortality after adjustment for co-morbidities, though the relation was attenuated. However, the association was no more present when NT-proBNP was added in the final model.

### Obesity paradox in HFrEF

Our results add to the demonstration that the relation between higher BMI and lower mortality in HFrEF is confounded by age, co-morbidities, and severity of disease as assessed by natriuretic peptides levels. In a recent metanalysis including independent patient data of 5819 heart-failure patients with predominantly reduced EF, Marcks et al. found that the obesity paradox was largely confined to elderly patients with co-morbidities, and disappeared after adjustment for NT-proBNP and troponin [17]. In a secondary analysis of the population included in PARADIGM, the apparent survival benefit in patients

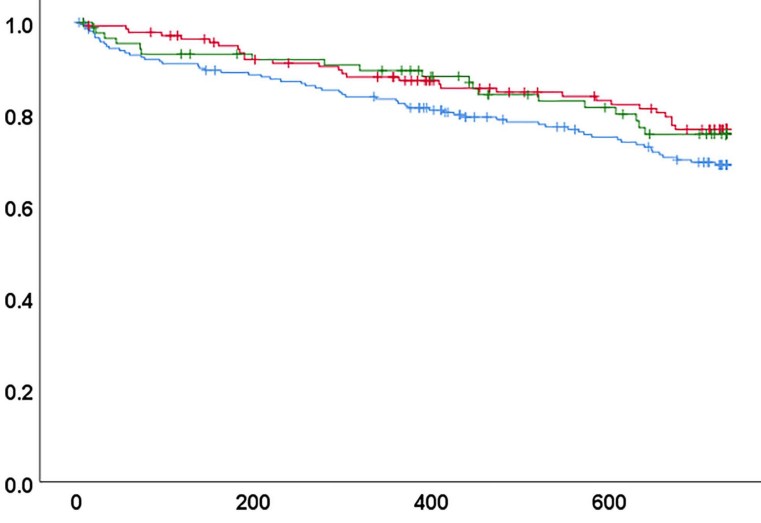

| Number at risk | 0 days | 200 d | 400 d | 600 d |
|---|---|---|---|---|
| Normal weight | 214 | 185 | 163 | 137 |
| Overweight | 140 | 123 | 105 | 93 |
| Obese | 89 | 77 | 69 | 57 |

**Fig 2. Survival by BMI categories, heart failure with reduced ejection fraction.** *(a)* p = 0.20 (logrank). Blue: normal weight (BMI 18.5-24.9). Red: overweight (BMI 25-29.9). Green: obese (BMI >= 30). X axis: days since inclusion. Y axis: survival without death.

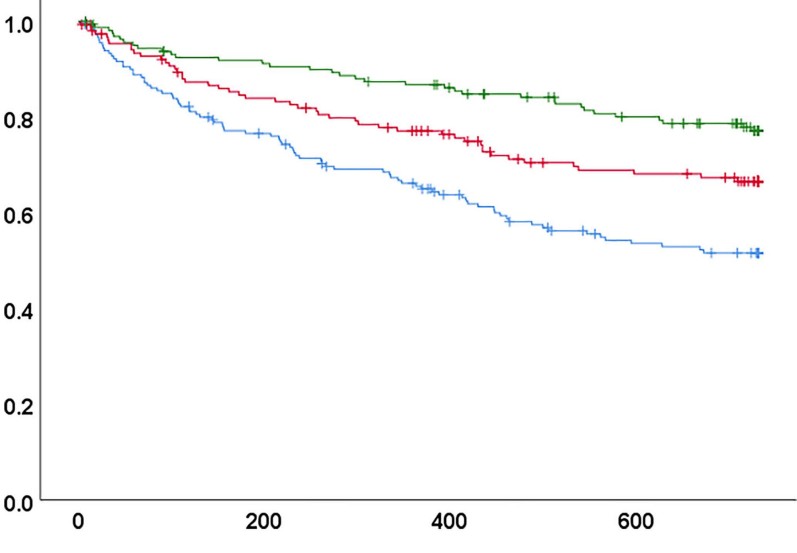

| Number at risk | 0 days | 200 d | 400 d | 600 d |
|---|---|---|---|---|
| Normal weight | 183 | 134 | 103 | 81 |
| Overweight | 155 | 124 | 105 | 88 |
| Obese | 162 | 144 | 132 | 116 |

**Fig 3. Survival by BMI categories, heart failure with preserved ejection fraction. p < 0.001 (logrank).** Blue: normal weight (BMI 18.5-24.9). Red: overweight (BMI 25-29.9). Green: obese (BMI >= 30). X axis: days since inclusion. Y axis: survival without death.

**Table 4. Two-years hazard ratio for all-cause death stratified by category of heart failure.**

| Co-variate | HR or aHR (95%CI) | p value | HR or aHR (95%CI) | p value |
|---|---|---|---|---|
| | HFpEF (n = 500) | | HFrEF (N = 443) | |
| **Unadjusted** | | | | |
| BMI (kg/m2) | 0.94 (0.92-0.97) | <0.001 | 0.97 (0.94-1.02) | 0.22 |
| **First model** | | | | |
| BMI (kg/m2) | 0.95 (0.91-0.99) | 0.02 | 0.99 (0.95-1.04) | 0.71 |
| Age (year) | 1.02 (1.00-1.05) | 0.24 | 1.03 (1.01-1.05) | 0.02 |
| Sex (ref: female) | 1.00 (0.70-1.38) | 0.99 | 0.78 (0.50-1.27) | 0.28 |
| Diabetes | 1.09 (0.73-1.65) | 0.65 | 1.01 (0.64-1.54) | 0.95 |
| Hypertension | 0.71 (0.47-1.13) | 0.10 | 0.95 (0.59-1.68) | 0.85 |
| Chronic obstructive pulmonary disease | 2.32 (1.55-3.48) | <0.01 | 1.82 (1.07-2.99) | 0.02 |
| Atrial fibrillation | 1.08 (0.75-1.52) | 0.66 | 1.35 (0.90-2.08) | 0.16 |
| Chronic anaemia | 1.75 (1.22-2.59) | <0.01 | 1.25 (0.80-1.98) | 0.33 |
| GFR (ml/mn) | 0.99 (0.98-1.00) | 0.11 | 0.99 (0.97-1.00) | 0.07 |
| **Second model** | | | | |
| BMI (kg/m2) | 0.98 (0.94-1.01) | 0.26 | 1.02 (0.96-1.07) | 0.59 |
| Age (year) | 1.02 (1.00-1.05) | 0.18 | 1.03 (1.01-1.06) | <0.01 |
| Sex (ref: female) | 0.96 (0.68-1.35) | 0.84 | 0.73 (0.47-1.29) | 0.32 |
| Diabetes | 1.13 (0.73-1.72) | 0.58 | 1.08 (0.65-1.72) | 0.73 |
| Hypertension | 0.68 (0.46-1.10) | 0.08 | 0.85 (0.52-1.55) | 0.54 |
| Chronic obstructive pulmonary disease | 2.23 (1.47-3.57) | <0.01 | 1.91 (1.02-3.38) | 0.02 |
| Atrial fibrillation | 1.02 (0.69-1.50) | 0.91 | 1.46(0.97-2.23) | 0.07 |
| Chronic anaemia | 1.62 (1.14-2.43) | <0.01 | 1.23 (0.78-1.93) | 0.36 |
| GFR (ml/mn) | 1.00 (0.99-1.01) | 0.64 | 0.99 (0.98-1.01) | 0.42 |
| NT-proBNP (log-transformed) | 2.61(1.62-4.34) | <0.01 | 2.64(1.50-4.99) | <0.01 |

with higher BMI disappeared after adjustment for other prognostic variables, including NT-proBNP [18]. This suggests that overweight patients with HFrEF have less advanced heart failure and catabolic state.

## Obesity paradox in HFpEF

The obesity paradox in HFpEF persisted after adjustment for age and co-morbidities. In a secondary analysis of the 3320 patients included in the TOPCAT trial, all suffering from HFpEF, Tsujimoto et al. found that the adjusted HR of 3-years death was 0.53 in obese vs. normoweight patients after extensive adjustment for confounding factors, including abdominal obesity [19]. However, they did not adjust for natriuretic peptides level. In our cohort, adding NT-proBNP in the multivariate model led to complete disappearance of any mortality benefit associated with excess weight.

Natriuretic peptides are affected, in addition to cardiac wall stress, by age, presence of atrial fibrillation, renal function, or obesity [20,21]. The relation between excess adipose tissue, heart failure severity, and natriuretic peptides is complex and not fully elucidated. Obese patients with heart failure, both HFrEF and HFpEF, have lower levels of natriuretic peptides than leaner patients [22,23]. This may partly reflect a less advanced disease, due to obese patients becoming symptomatic earlier (lead time bias). Another explanation is altered secretion, clearance, or metabolism of natriuretic peptides mediated by the adipose tissue [24]. Indeed, a lower level of natriuretic peptides is also observed in obese healthy subjects [25]. Biological hypothetical explanations include increased expression of neprilysin by the adipose tissue (resulting in higher clearance of natriuretic peptides), or higher insulinemia leading to inactivation of natriuretic peptides by insulin degrading enzyme.[26] Natriuretic peptide clearance receptors-C are also highly expressed in adipose tissue, leading to

lower level of BNP but not NT-proBNP [27]. Finally, the relation is bidirectional, as natriuretic peptides strongly activate lipolysis, which can amplify the inverse association between excess adipose tissue and low natriuretic peptides levels [28]. S2 Fig Natriuretic peptides are still predictive of mortality in obese patients with heart failure [23].

Patients with HFpEF and obesity may have a different physiopathology than lean patients with HFpEF. Hyperaldosteronism, activation of the sympathetic system, increased leptin and neprilysin secretion by adipocytes are characteristics of obesity, and result in increased circulating blood volume, decreased ventricular compliance, and increased inflammation [29–31]. Inflammation associated with ageing and common co-morbidities is characteristic of HFpEF, and obesity by itself promotes an inflammatory milieu [31]. C-reactive protein, the circulating biomarker of inflammation available in our study, was elevated in all categories of BMI and tended to rise with increasing BMI. The elevated C-reactive level in our cohort is probably related to the hospital setting, with many admitted patients suffering from multiple conditions, including COPD or concomitant infections.

Our findings highlight that the apparent survival benefit of overweight or obese patients with heart failure is confounded by younger age, less comorbidities, and less advanced heart failure, but this demonstration requires extensive adjustment including natriuretic peptides levels. This is of particular importance and relevance so as not to hinder weight loss promotion in this population for fear of negatively impacting their prognosis.

GLP-1 agonists are promising in the treatment of HFpEF and induce significant weight loss. In the STEP HFpEF trial, one year of semaglutide administered to HFpEF patients with a BMI > 30 kg/m$^2$ led to a mean percentage loss in body weight of 10.7%, improvement in symptoms of heart failure and in 6-minutes' walk distance compared to placebo. Heart-failure events, though rare, were more frequent with placebo. Similar results were apparent in the STEP-HFpEF DM trial, which included HFpEF patients with obesity and diabetes [13,14]. In the SUMMIT trial, 731 patients with HFpEF and obesity were randomized to one year of tirzepatide, a dual agonist of glucose-dependent insulinotropic polypeptide and glucagon-like peptide-1 receptor. The composite primary endpoint of death (cardiovascular causes or worsening heart-failure) was lower in the tirzepatide arm. The mean percent loss in body weight was 11.6% higher in the tirzepatide arm [15]. The results of these trials suggest that pharmacologically induced weight loss can be safely achieved with concomitant improvement in heart failure-related events. A recent analysis of real-world cohorts emulating the design of the STEP-HFpEF and SUMMIT trials have described congruent findings, with marked reduction in heart failure-related hospitalization or overall mortality [32]. Though these data are encouraging, part of the beneficial effects of GLP-1 agonists could be independent of weight loss, and these results cannot be directly extrapolated to weight loss resulting from lifestyle interventions in heart failure [33].

Due to its easy availability, BMI is widely used as an anthropometric correlate of obesity. However, BMI does not account for the relative part of lean mass, fat mass, and bones in body composition, nor for the distribution of adipose tissue (subcutaneous or visceral). The waist-to-height ratio better reflects visceral obesity, and the obesity paradox is not apparent when using this surrogate of obesity in patients with HFrEF.[18] Waist-to-height ratio should be used as the preferential anthropometric measure of obesity in future trials investigating the prognosis of obesity in heart failure [34]. Epicardial adipose tissue is tightly involved with myocardial function, metabolism, and hemodynamics.[35] Moreover, its effect may differ between HFpEF and HFrEF phenotypes, hence providing a mechanism by which the excess of adipose tissue may differentially affect overweight patients [35]. Direct measures of epicardial adipose tissue were however not available in our cohort.

Inclusion of natriuretic peptides in the diagnostic pathway of heart failure leads to substantial increase in accuracy and is widely advocated.[11,20]. Age-adjusted cut-offs for the diagnosis of heart failure have been proposed to improve the specificity of NT-proBNP [36]. Conversely, the intrinsically lower level of natriuretic peptides in obese patients may lower the sensitivity of established rule-out cut-offs in obese patients [24]. We chose not to use age-adjusted cut-offs for inclusion of our patients, as it would lead to selective exclusion of patients in the obese and overweight categories. Natriuretic peptides are also closely associated with the prognosis of both HFrEF and HFpEF including in obese patients justifying their inclusion as a confounder in the multivariate model [23,37–40]

Among the strength of our findings are the enrolment of an unselected population of patients hospitalized for acute heart failure during five years. The prospective design, and strict verification of clinical, echocardiographic, and biologic characteristics lead to a low risk of misclassification or overdiagnosis of heart failure. Multiple characteristics of all patients were collected and there were few missing data, allowing for extensive adjustment for confounding factors.

### Study limitations

Our work has some limitations. Patients with grade II or III obesity were a minority, and our findings apply to the whole spectrum of overweight, mild, moderate and severe obesity. Weight and height were obtained from routine clinical practice and were not standardized, and inaccuracies are possible. This is especially true for height, that was not measured but self-reported which may introduce bias in BMI calculation. However, errors should be minor or at random, so they are unlikely to affect the conclusions. We relied on BMI as a surrogate for obesity, although an elevated BMI not always reflects an excess of adipose tissue. However, other biometric measures (waist-to-height ratio, body composition by bio-impedance) were not available. Also, the relationship between visceral or subcutaneous fat and prognosis might differ, and we had no indication of predominant fat repartition in our patients. To account for lower levels of natriuretic peptides in obese patients, lowering the diagnostic cut-off by up to 40% in obese patients has been recently proposed [20,36]. Such adjustment is based on expert opinion and still has to prove its usefulness. However, using a fixed cut-off as in the present study might lead to inclusion of patients with more severe disease in the obese category.

It was a single-centre study concerning exclusively hospitalized patients, and the generalization of our findings to the ambulatory setting or to other healthcare systems should be made with caution. Lack of data on the weight trajectory of the patients before enrolment in the registry prevented us to account for time-varying exposure to high BMI. Finally, we lacked information on other important variables like cardiorespiratory fitness or socioeconomic factors, so residual confounding cannot be excluded.

### Conclusions

The seemingly better prognosis conferred by increased BMI in patients with heart failure disappears after extensive adjustment for co-morbidities and NT-proBNP. The mortality benefit associated with increased BMI (obesity paradox) can be fully explained by differences in age, comorbidities and severity of heart failure.

### Supporting information

**S1 Fig. Survival by BMI categories, all patients including underweight category.**
(TIF)

**S2 Fig. Causal framework for the relation between adipose tissue, natriuretic peptides, and mortality.**
(TIF)

**S1 File. Database_PLOSONE_2026.**
(XLSX)

### Author contributions

**Conceptualization:** Nicolas Garin.

**Formal analysis:** Nicolas Garin.

**Investigation:** David Carballo, Sebastian Carballo.

**Project administration:** Sebastian Carballo.

**Supervision:** Sebastian Carballo.

**Writing – original draft:** Nicolas Garin.

**Writing – review & editing:** Nicolas Garin, David Carballo, Jonathan Dash, Jérôme Stirnemann, Jean-Luc Reny, Nicolas Vuilleumier, Sebastian Carballo.

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
