## [Decision Letter · Decision Letter 0]

29 Oct 2025

Dear Dr. Garin,

We look forward to receiving your revised manuscript.

Kind regards,

Yoshiaki Taniyama, MD, PhD

Academic Editor

PLOS ONE

Journal Requirements:

3. Please upload a new copy of Figures 1A, 1B, and 1C as the details are not clear. Please follow the link for more information: https://journals.plos.org/plosone/s/figures

Reviewers' comments:

Reviewer's Responses to Questions

**Comments to the Author**

1. Is the manuscript technically sound, and do the data support the conclusions?

Reviewer #1: Yes

Reviewer #2: Yes

2. Has the statistical analysis been performed appropriately and rigorously?

Reviewer #1: Yes

Reviewer #2: Yes

3. Have the authors made all data underlying the findings in their manuscript fully available?

Reviewer #1: Yes

Reviewer #2: Yes

4. Is the manuscript presented in an intelligible fashion and written in standard English?

Reviewer #1: Yes

Reviewer #2: Yes

Reviewer #1: The manuscript addresses the so-called “obesity paradox” in patients hospitalized with acute heart failure. The study is timely, clinically relevant, and based on a large, prospectively collected cohort with detailed characterization and long-term follow-up. The manuscript is clearly written and methodologically sound. Nevertheless, I have several concerns that should be addressed before the manuscript can be considered for publication.

Major comments

1. Definition and distribution of obesity

Patients with BMI ≥30 kg/m² were grouped into a single “obese” category. However, Table 1 shows that the mean BMI in this group was 34.7 (SD 4.3) kg/m², suggesting that most patients had only mild obesity (class I). Please provide the median and interquartile range of BMI within the obese subgroup. If possible, stratify the obese category into class I vs. class II–III obesity. This would test whether the “paradox” truly applies to patients with more severe obesity.

The Discussion and Conclusions should clearly state that the findings mainly apply to patients with overweight or mild obesity, while extrapolation to severe obesity remains at least uncertain.

2. Model complexity and risk of overfitting

While I appreciate the clarity of the analyses and the consistency of the results across sensitivity models, I remain concerned about the possibility of model overfitting, especially in the subgroup analyses where the number of events is more limited (e.g., HFrEF). Please confirm whether proportional hazards assumptions were formally tested for the Cox models.

I would also kindly suggest that the authors strengthen their results by including internal validation procedures. In particular, bootstrap resampling (e.g., 1000 repetitions) could be used to quantify optimism-corrected performance measures such as Harrell’s C-index, Brier score at 2 years, integrated Brier score, and calibration slope. Calibration plots (apparent vs. optimism-corrected) would also be informative. Reporting these metrics separately for the overall cohort and for HFpEF/HFrEF subgroups would provide reassurance that the findings are not driven by model instability.

If the optimism-corrected calibration slope is found to be <1, a global shrinkage factor or penalized Cox regression (ridge) could be considered as a sensitivity analysis.

3. Adjustment for NT-proBNP

The attenuation of the association between BMI and mortality after adjusting for NT-proBNP is a crucial finding. In the overall cohort, the adjusted HR was 0.99 (95% CI 0.96–1.01), i.e., no significant association. This should be highlighted, as the manuscript currently emphasizes the persistence of the paradox in HFpEF, but the global adjusted analysis is essentially negative.

NT-proBNP is strongly influenced by adiposity. It may act as both a confounder and a mediator. This issue deserves explicit discussion, ideally with a causal framework.

In addition, considering alternative functional forms for NT-proBNP (e.g., log-transformation or low-degree splines) could further exclude residual model misspecification.

4. Residual confounding and limitations of BMI

BMI is a crude surrogate for adiposity and does not differentiate between lean and fat mass, nor does it account for fat distribution. Evidence suggests that the “paradox” attenuates when using waist-to-height ratio or direct measures of body composition (please cite 10.1186/S12933-025-02778-6).

Also, C-reactive protein (CRP), the available marker of systemic inflammation, was not significantly associated with BMI in either HFpEF or HFrEF (Tables 3–4). Given that obesity is commonly linked to low-grade inflammation (please cite 10.1093/cvr/cvac133), I think it would be important for the authors to comment on this apparent lack of association. A brief discussion of potential explanations (e.g., confounding, limited sensitivity of CRP, or phenotype-specific mechanisms) would add clarity and context to the findings.

Minor comments

1. In the Methods, clarify that height was self-reported, which may have introduced bias in BMI calculation.

2. Figures 1b and 1c: axis labels should use larger fonts to improve readability. Adding numbers at risk below the Kaplan–Meier curves would also help interpretation.

Reviewer #2: This manuscript investigates the association between body mass index (BMI) and mortality among patients hospitalized for acute heart failure (HF), with separate analyses for HF with preserved (HFpEF) and reduced ejection fraction (HFrEF). The authors report that a higher BMI is only associated with better survival only in HFpEF patients, even after multivariable adjustment including NT-proBNP. This topic is clinically and pathophysiologically relevant, especially given the increasing prevalence of obesity and the emerging use of weight-lowering agents in HF management. The study is generally well-written and based on a well-defined cohort. It contributes to our understanding of the heterogeneity of the “obesity paradox” in HF phenotypes.

However, several methodological and interpretative issues should be addressed before the manuscript can be considered for publication.

Major Comments

1. Adjustment Strategy for NT-proBNP

The inclusion of NT-proBNP in the multivariable model is appropriate; however, since its distribution is typically exponential (right-skewed), it should be log-transformed prior to analysis.

Additionally, presenting sensitivity models in which NT-proBNP is included as a categorical variable (e.g., tertiles) would further demonstrate the robustness of the findings.

2. Heart Failure Phenotype Classification

Patients with “mildly reduced EF (41–49%)” were merged into the HFrEF group. Given recent ESC and AHA classifications that consider HFmrEF as a separate phenotype, a sensitivity analysis excluding or separately analyzing these patients is recommended to confirm that this choice does not influence the results.

3. Nutritional Status Across BMI Categories

While BMI is a crude measure of obesity, it does not directly reflect nutritional status. It would strengthen the study to include data (if available) on serum albumin, total cholesterol, or other nutritional indices (e.g., CONUT or PNI score) across BMI groups. This would clarify whether the observed association reflects true adiposity or better nutritional reserve.

4. Distribution of Heart Failure Medications

Pharmacologic treatment significantly influences prognosis in HF. Please provide the distribution of key heart failure therapies (e.g., ACEI/ARB/ARNI, β-blockers, MRA, SGLT2 inhibitors) across BMI categories and between HFpEF and HFrEF. Adjustment for medication use, at least in sensitivity analyses, would improve the validity of the observed associations.

5. Underweight Group Exclusion

Table 2 and Figure 1 exclude the underweight group, yet their mortality is of interest given their potential frailty and catabolic state. Please provide descriptive and outcome data for this subgroup (even if excluded from multivariable models), or clearly explain the rationale for exclusion from survival analysis. Inclusion of their Kaplan–Meier curve or summary results in supplementary materials would be helpful.

6. Potential Selection and Residual Confounding

As this registry includes only hospitalized HF patients, selection bias toward more severe cases may exist. Discuss whether admission criteria or disease severity at presentation could differ by BMI category.

Additionally, residual confounding by unmeasured variables—such as inflammation, cardiorespiratory fitness, or socioeconomic factors—should be discussed as alternative explanations of the observed “obesity paradox.”

7. Statistical Power and Interaction Testing

The reported interaction between BMI and HF phenotype (p = 0.02) suggests heterogeneity, but subgroup sample sizes may limit statistical power. Please provide the number of events per variable (EPV) in each model and discuss whether this interaction was adequately powered.

Minor Comments

1. Tables and Figures

The legends in Figure 1 are duplicated; please correct them appropriately.

Figure 1 would benefit from including the number at risk at the bottom of each Kaplan–Meier curve.

In Table 1, “Rales: moderate or marked” might be replaced with “Pulmonary rales (≥ moderate)” for clarity.

**Do you want your identity to be public for this peer review?** For information about this choice, including consent withdrawal, please see our Privacy Policy

Reviewer #1: No

Reviewer #2: No

---

## [Author Response · Author response to Decision Letter 1]

4 Dec 2025

We would like to sincerely thank both reviewers for their genuine interest in this manuscript. Specifically, their comments allowed to strengthen the analyses, which led to a major change in the interpretation of the data. Indeed, BMI lost all association with mortality when adjusting for NT-proBNP, both for patients with reduced and with preserved ejection fraction. This highlights that the so-called obesity paradox is mainly due to confusion, and that previous studies that did not adjust for natriuretic peptides could not entirely deal with residual confusion.

Further comments of the reviewers allowed more detailed discussion of the relationships between adipose tissue and HFpEF, specially regarding the effects of GLP agonists in HF.

Reviewer #1: The manuscript addresses the so-called “obesity paradox” in patients hospitalized with acute heart failure. The study is timely, clinically relevant, and based on a large, prospectively collected cohort with detailed characterization and long-term follow-up. The manuscript is clearly written and methodologically sound. Nevertheless, I have several concerns that should be addressed before the manuscript can be considered for publication.

Thank you for this supporting comment

Major comments

1. Definition and distribution of obesity

Patients with BMI ≥30 kg/m² were grouped into a single “obese” category. However, Table 1 shows that the mean BMI in this group was 34.7 (SD 4.3) kg/m², suggesting that most patients had only mild obesity (class I). Please provide the median and interquartile range of BMI within the obese subgroup. If possible, stratify the obese category into class I vs. class II–III obesity. This would test whether the “paradox” truly applies to patients with more severe obesity.

Median BMI in the obese category was 34 kg/ m2 (IQR 32-37); among the 256 obese patients, 144 had class I and 112 class II or III obesity. Overall and heart failure related mortality decreased according to the class of obesity, confirming that the obesity paradox also applies to severe obesity. This stratification has been added in the method section and the corresponding results have been introduced in the Result section and in Table 2

The Discussion and Conclusions should clearly state that the findings mainly apply to patients with overweight or mild obesity, while extrapolation to severe obesity remains at least uncertain.

Thank you for this relevant comment. Actually, our findings are not restrained to obesity but also concern overweight patients.

We first modified the Title accordingly, that now reads:

“Overweight and obesity association with mortality in patients with heart failure and reduced or preserved ejection fraction-a cohort study «

We acknowledge in the limitations that patients with moderate or severe obesity formed a minority of the included population.

“Patients with grade II or III obesity were a minority and our findings apply to the whole spectrum of overweight, mild, moderate and severe obesity »

2. Model complexity and risk of overfitting

While I appreciate the clarity of the analyses and the consistency of the results across sensitivity models, I remain concerned about the possibility of model overfitting, especially in the subgroup analyses where the number of events is more limited (e.g., HFrEF). Please confirm whether proportional hazards assumptions were formally tested for the Cox models.

I would also kindly suggest that the authors strengthen their results by including internal validation procedures. In particular, bootstrap resampling (e.g., 1000 repetitions) could be used to quantify optimism-corrected performance measures such as Harrell’s C-index, Brier score at 2 years, integrated Brier score, and calibration slope. Calibration plots (apparent vs. optimism-corrected) would also be informative. Reporting these metrics separately for the overall cohort and for HFpEF/HFrEF subgroups would provide reassurance that the findings are not driven by model instability.

If the optimism-corrected calibration slope is found to be <1, a global shrinkage factor or penalized Cox regression (ridge) could be considered as a sensitivity analysis.

Thank you for this insightful comment. The proportional hazards assumption was assessed by direct examination of the log-minus-log plots. The assumption was met. To strengthen confidence in the results of the multivariate analysis, we now present robust 95% CI obtained with bootstrapping.

These points have been added in the Methods and Results sections of the manuscript.

3. Adjustment for NT-proBNP

The attenuation of the association between BMI and mortality after adjusting for NT-proBNP is a crucial finding. In the overall cohort, the adjusted HR was 0.99 (95% CI 0.96–1.01), i.e., no significant association. This should be highlighted, as the manuscript currently emphasizes the persistence of the paradox in HFpEF, but the global adjusted analysis is essentially negative.

NT-proBNP is strongly influenced by adiposity. It may act as both a confounder and a mediator. This issue deserves explicit discussion, ideally with a causal framework.

In addition, considering alternative functional forms for NT-proBNP (e.g., log-transformation or low-degree splines) could further exclude residual model misspecification.

Thank you for this comment. Appropriate modelling of the data is indeed a crucial issue.

NT-proBNP distribution in this study was right-skewed, and log transformation allowed effective normalization of the distribution (see below).

NT-proBNP distribution

NT-proBNP distribution after log transformation

When introducing log transformed NT-proBNP in the fully adjusted model (Model 2), the association between BMI and HR of mortality disappeared completely, both for HFpEF and HFrEF patients. The conclusion is that the obesity paradox is indeed fully explained in our cohort by confounding, and that full adjustment including natriuretic peptides is required to control for differences in severity of heart failure between overweight / obese patients and their leaner counterparts.

These findings are more consistent with recent literature results and concepts and lead to major changes in the results and discussion section, as well as in the title and in the abstract. We now state in the Discussion section:

“Our findings highlight that the apparent survival benefit of overweight or obese patients with heart failure is confounded by younger age, less comorbidities, and less advanced heart failure, but this demonstration requires extensive adjustment including for differences in natriuretic peptides levels.”

The Conclusion now reads:

“The seemingly better prognosis conferred by increased BMI in patients with heart failure disappears after extensive adjustment for co-morbidities and NT-proBNP. The mortality benefit associated with increased BMI (obesity paradox) can be fully explained by differences in age, comorbidities and severity of heart failure.”

As for the complex relation between natriuretic peptides and obesity, it is extensively discussed in the Discussion section. We added a causal framework as a supplement figure.

4. Residual confounding and limitations of BMI

BMI is a crude surrogate for adiposity and does not differentiate between lean and fat mass, nor does it account for fat distribution. Evidence suggests that the “paradox” attenuates when using waist-to-height ratio or direct measures of body composition (please cite 10.1186/S12933-025-02778-6).

Thank you for raising this crucial point. Unfortunately, no other biometric measure of adiposity was available in our cohort. We extended the discussion section with a paragraph acknowledging the importance of epicardial adipose tissue measurement and its relations with myocardial function.

Also, C-reactive protein (CRP), the available marker of systemic inflammation, was not significantly associated with BMI in either HFpEF or HFrEF (Tables 3–4). Given that obesity is commonly linked to low-grade inflammation (please cite 10.1093/cvr/cvac133), I think it would be important for the authors to comment on this apparent lack of association. A brief discussion of potential explanations (e.g., confounding, limited sensitivity of CRP, or phenotype-specific mechanisms) would add clarity and context to the findings.

Thank you for this important ref that is now cited in the discussion section. We also comment on the trend towards increased CRP levels with increased BMI, and to the elevated CRP level in our cohort.

“Inflammation associated with ageing and common co-morbidities is characteristic of HFpEF, and obesity by itself promotes an inflammatory milieu. Ref C-reactive protein, the circulating biomarker of inflammation available in our study, was elevated in all categories of BMI and tended to rise with increasing BMI. The elevated C-reactive level in our cohort is probably related to the hospital setting, with many admitted patients suffering from multiple conditions, including COPD or concomitant infections.”

Minor comments

1. In the Methods, clarify that height was self-reported, which may have introduced bias in BMI calculation.

We acknowledge this limitation; the corresponding paragraph in the discussion section now reads:

“Weight and height were obtained from routine clinical practice and were not standardized, and inaccuracies are possible. This is especially true for height, that was not measured but self-reported which may introduce bias in BMI calculation.”

2. Figures 1b and 1c: axis labels should use larger fonts to improve readability. Adding numbers at risk below the Kaplan–Meier curves would also help interpretation.

Done

Reviewer #2: This manuscript investigates the association between body mass index (BMI) and mortality among patients hospitalized for acute heart failure (HF), with separate analyses for HF with preserved (HFpEF) and reduced ejection fraction (HFrEF). The authors report that a higher BMI is only associated with better survival only in HFpEF patients, even after multivariable adjustment including NT-proBNP. This topic is clinically and pathophysiologically relevant, especially given the increasing prevalence of obesity and the emerging use of weight-lowering agents in HF management. The study is generally well-written and based on a well-defined cohort. It contributes to our understanding of the heterogeneity of the “obesity paradox” in HF phenotypes.

Again, many thanks for this encouraging comment.

However, several methodological and interpretative issues should be addressed before the manuscript can be considered for publication.

Major Comments

1. Adjustment Strategy for NT-proBNP

The inclusion of NT-proBNP in the multivariable model is appropriate; however, since its distribution is typically exponential (right-skewed), it should be log-transformed prior to analysis.

Additionally, presenting sensitivity models in which NT-proBNP is included as a categorical variable (e.g., tertiles) would further demonstrate the robustness of the findings.

Thank you for this major point that led to significant changes in the results of the fully adjusted model. See also response to reviewer 1.

2. Heart Failure Phenotype Classification

Patients with “mildly reduced EF (41–49%)” were merged into the HFrEF group. Given recent ESC and AHA classifications that consider HFmrEF as a separate phenotype, a sensitivity analysis excluding or separately analyzing these patients is recommended to confirm that this choice does not influence the results.

As the results of the fully adjusted model are now homogeneous for both phenotypes of HF, this sensitivity analysis is not yet needed.

3. Nutritional Status Across BMI Categories

While BMI is a crude measure of obesity, it does not directly reflect nutritional status. It would strengthen the study to include data (if available) on serum albumin, total cholesterol, or other nutritional indices (e.g., CONUT or PNI score) across BMI groups. This would clarify whether the observed association reflects true adiposity or better nutritional reserve.

Only total cholesterol was available. Levels did not differ significantly between BMI categories. Results were added in Table 1

4. Distribution of Heart Failure Medications

Pharmacologic treatment significantly influences prognosis in HF. Please provide the distribution of key heart failure therapies (e.g., ACEI/ARB/ARNI, β-blockers, MRA, SGLT2 inhibitors) across BMI categories and between HFpEF and HFrEF. Adjustment for medication use, at least in sensitivity analyses, would improve the validity of the observed associations.

Thank you. Prescription of angiotensin converting enzyme inhibitors, angiotensin receptor blockers or mineralocorticoid receptor antagonists was not significantly different between weight categories. The corresponding results have been added to Table 1 and in the Results section. Indeed, these medications modify the prognosis of the disease only in patients with HFrEF. SGLT-2i are effective both in patients with HFpEF and HFrEF. However, when the patients of the cohort were included, SGLT-2i were not yet largely used.

5. Underweight Group Exclusion

Table 2 and Figure 1 exclude the underweight group, yet their mortality is of interest given their potential frailty and catabolic state. Please provide descriptive and outcome data for this subgroup (even if excluded from multivariable models), or clearly explain the rationale for exclusion from survival analysis. Inclusion of their Kaplan–Meier curve or summary results in supplementary materials would be helpful.

We agree that this subgroup is of special interest, as it certainly includes patients with more advanced heart failure characterized by a catabolic state. However, our data don’t allow to exclude that some of them might be affected by cancer or another chronic inflammatory illness, which can confer them a worse diagnosis.

According to the suggestion, we added outcome data of these patients in Table 2 and provide a Kaplan Meier analysis including this category as a supplementary file.

6. Potential Selection and Residual Confounding

As this registry includes only hospitalized HF patients, selection bias toward more severe cases may exist. Discuss whether admission criteria or disease severity at presentation could differ by BMI category.

Additionally, residual confounding by unmeasured variables—such as inflammation, cardiorespiratory fitness, or socioeconomic factors—should be discussed as alternative explanations of the observed “obesity paradox.”

Indeed, the obesity paradox has completely disappeared when log transformation to NT-proBNP levels was applied. The following statement was added to the Study limitations:

“Finally, we lacked information on other important variables like cardiorespiratory fitness or socioeconomic factors, so residual confounding cannot be excluded.”

7. Statistical Power and Interaction Testing

The reported interaction between BMI and HF phenotype (p = 0.02) suggests heterogeneity, but subgroup sample sizes may limit statistical power. Please provide the number of events per variable (EPV) in each model and discuss whether this interaction was adequately powered.

We suppressed this interaction test as the results of the analyses stratified by HF phenotype are now congruent

Minor Comments

1. Tables and Figures

The legends in Figure 1 are duplicated; please correct them appropriately.

Done

Figure 1 would benefit from including the number at risk at the bottom of each Kaplan–Meier curve.

Done

In Table 1, “Rales: moderate or marked” might be replaced with “Pulmonary rales (≥ moderate)” for clarity.

Done

---

## [Decision Letter · Decision Letter 1]

11 Jan 2026

Overweight and obesity association with mortality in patients with heart failure and reduced or preserved ejection fraction-a cohort study

PONE-D-25-37278R1

Dear Dr. Garin,

We’re pleased to inform you that your manuscript has been judged scientifically suitable for publication and will be formally accepted for publication once it meets all outstanding technical requirements.

Kind regards,

Yoshiaki Taniyama, MD, PhD

Academic Editor

PLOS One

Additional Editor Comments (optional):

Reviewers' comments:

Reviewer's Responses to Questions

**Comments to the Author**

Reviewer #1: All comments have been addressed

2. Is the manuscript technically sound, and do the data support the conclusions?

Reviewer #1: Yes

3. Has the statistical analysis been performed appropriately and rigorously?

Reviewer #1: Yes

4. Have the authors made all data underlying the findings in their manuscript fully available?

Reviewer #1: Yes

5. Is the manuscript presented in an intelligible fashion and written in standard English?

Reviewer #1: Yes

Reviewer #1: The manuscript addresses the so-called “obesity paradox” in patients

hospitalized with acute heart failure. The study is timely, clinically relevant, and based

on a large, prospectively collected cohort with detailed characterization and long-term

follow-up. The manuscript is clearly written and methodologically sound. The authors answered my previous comments..........

**Do you want your identity to be public for this peer review?** For information about this choice, including consent withdrawal, please see our Privacy Policy

Reviewer #1: No

---

## [Editor Report · Acceptance letter]

PONE-D-25-37278R1

PLOS One

Dear Dr. Garin,

I'm pleased to inform you that your manuscript has been deemed suitable for publication in PLOS One. Congratulations! Your manuscript is now being handed over to our production team.

Kind regards,

on behalf of

Dr. Yoshiaki Taniyama

Academic Editor

PLOS One